# Nutrition Profile for Countries of the Eastern Mediterranean Region with Different Income Levels: An Analytical Review

**DOI:** 10.3390/children10020236

**Published:** 2023-01-28

**Authors:** Hanna Leppäniemi, Eman T. Ibrahim, Marwa M. S. Abbass, Elaine Borghi, Monica C. Flores-Urrutia, Elisa Dominguez Muriel, Giovanna Gatica-Domínguez, Richard Kumapley, Asmus Hammerich, Ayoub Al-Jawaldeh

**Affiliations:** 1Regional Office for the Eastern Mediterranean (EMRO), World Health Organization (WHO), Cairo 7608, Egypt; leppaniemih@who.int (H.L.); eibrahim@who.int (E.T.I.); hammericha@who.int (A.H.); aljawaldeha@who.int (A.A.-J.); 2Oral Biology Department, Faculty of Dentistry, Cairo University, Cairo 11553, Egypt; 3Department of Nutrition and Food Safety, World Health Organization, 1211 Geneva, Switzerland; borghie@who.int (E.B.); floresm@who.int (M.C.F.-U.); domingueze@who.int (E.D.M.); gaticag@who.int (G.G.-D.); kumapleyr@who.int (R.K.)

**Keywords:** nutrition profiles, EMR, income groups, child malnutrition

## Abstract

The World Health Organization’s (WHO) Eastern Mediterranean Region (EMR) is suffering from a double burden of malnutrition in which undernutrition coexists with rising rates of overweight and obesity. Although the countries of the EMR vary greatly in terms of income level, living conditions and health challenges, the nutrition status is often discussed only by using either regional or country-specific estimates. This analytical review studies the nutrition situation of the EMR during the past 20 years by dividing the region into four groups based on their income level—the low-income group (Afghanistan, Somalia, Sudan, Syria, and Yemen), the lower-middle-income group (Djibouti, Egypt, Iran, Morocco, Pakistan, Palestine, and Tunisia), the upper-middle-income group (Iraq, Jordan, Lebanon, and Libya) and the high-income group (Bahrain, Kuwait, Oman, Qatar, Saudi Arabia, and United Arab Emirates)—and by comparing and describing the estimates of the most important nutrition indicators, including stunting, wasting, overweight, obesity, anaemia, and early initiation and exclusive breastfeeding. The findings reveal that the trends of stunting and wasting were decreasing in all EMR income groups, while the percentages of overweight and obesity predominantly increased in all age groups across the income groups, with the only exception in the low-income group where a decreasing trend among children under five years existed. The income level was directly associated with the prevalence rates of overweight and obesity among other age groups except children under five, while an inverse association was observed regarding stunting and anaemia. Upper-middle-income country group showed the highest prevalence rate of overweight among children under five. Most countries of the EMR revealed below-desired rates of early initiation and exclusive breastfeeding. Changes in dietary patterns, nutrition transition, global and local crises, and nutrition policies are among the major explanatory factors for the findings. The scarcity of updated data remains a challenge in the region. Countries need support in filling the data gaps and implementing recommended policies and programmes to address the double burden of malnutrition.

## 1. Introduction

In the countries of the World Health Organization’s (WHO) Eastern Mediterranean Region (EMR), a complex combination of dietary practices alongside with environmental, social, and economic factors have resulted in the persistence of double burden of malnutrition, in which undernutrition coexists with rising rates of overweight, obesity, and diet-related chronic diseases [1]. The health of billions of people worldwide is at risk due to the rising prevalence of overweight and obesity. Over 1.9 billion persons aged 18 and older were overweight in 2016 and more than 600 million of these people were obese [2]. In 2020, almost 39 million children under five years of age were overweight—an increase of 5.6 million since 2000 [3]. High body mass index (BMI) accounted for an estimated four million deaths in 2015, mainly due to cardiovascular diseases (CVDs) [4]. In an extensive meta-analysis with 10,625,411 participants, all-cause mortality increased with higher BMI within people who were overweight or obese [5]. Overweight and obesity are risk factors for many noncommunicable diseases (NCDs) including CVDs, type 2 diabetes, and certain types of cancers. NCDs are the leading causes of death globally—as well as in the EMR—and were responsible for an estimated 41.1 million (73.4%) of the 55.9 million global deaths in 2017 [6,7].

Malnutrition is a major risk factor for death and disability [8]. Globally, roughly one in every three persons suffers from some type of malnutrition [9,10], and current trends predict that this number will rise to one in every two by 2025 [11]. Stunting, wasting, and micronutrient deficiencies are all examples of malnutrition which are caused by a lack of energy or nutrients. Both types of malnutrition in childhood, undernutrition as well as excessive and imbalanced consumption, are linked to adulthood obesity, overweight, and NCDs [12]. According to the latest global estimates, around 149 million (22%) children under the age of five are stunted, 45 million (6.7%) are wasted, and 39 million (5.7%) are overweight [3]. All groups are at an elevated risk of developing NCDs as adults [9]. Child and maternal malnutrition were responsible for 11.6% of disability-adjusted life years (DALYs) and 2.94 million deaths globally in 2019 [13].

Child and maternal malnutrition are the number one risk factor responsible for the most deaths and disabilities in six countries in the EMR (Afghanistan, Djibouti, Pakistan, Somalia, Sudan, and Yemen) according to the 2019 Global Burden of Disease Study. The highest increase in the burden attributable to malnutrition between 2009 and 2019 in the EMR was reported in Qatar (21.7%), followed by Oman (21%), and Pakistan (17%) [14].

Micronutrient deficiencies with associated health problems are some of the great malnutrition challenges in the EMR. Iron deficiency and anaemia associated with it are widespread among women, especially those of child-bearing age, and among children. Anaemia can result in fatigue, dizziness, reduced work capacity, loss of productivity, and increased susceptibility to infections [15,16,17]. During pregnancy, anemia increases the risk of serious health consequences for both the mother and the baby, including miscarriage, stillbirth, preterm delivery, intrauterine growth retardation, low birth weight, and mortality [18,19]. The WHO’s Global Health Observatory (GHO) reported a regional prevalence of anaemia in 2019 of 34.9% in women of reproductive age, and 42.7% in children 6–59 months [20]. According to a recent regional review article carried out by Al-Jawaldeh et al., anaemia prevalence in the countries of the EMR ranged from 27% to 69.6% among women of reproductive age and from 23.8% to 83.5% among children under five years [21], with similar trends observed at the WHO-GHO anaemia estimates [20]. The situation was most alarming in Yemen, where the prevalence of anaemia was 69.6% among women of reproductive age and 83.5% among children under five. Additionally, Somalia, Pakistan, and Afghanistan were identified to have anaemia as severe public health problem (prevalence exceeding 40%) [22] both among women of reproductive age and children under five [20,21].

The WHO-EMR comprises 22 countries with a total population of 730.524 million (725.721 million for the WHO-EMR countries and 4.803 million for West Bank and Gaza) in 2020 [23] (Figure 1). These 22 countries vary greatly in terms of income level, from the world’s highest to lowest. Moreover, the citizens of the EMR countries have unequal access to technology, water, sanitation, and electricity [24]. Because of this variety within the region regarding socioeconomic status (SES) and living conditions, the nutrition status of the region cannot be viewed only through regional averages which would not tell the whole truth of the diverse situations within the EMR. Additionally, the regional estimates are heavily influenced by the most populous countries such as Pakistan, Egypt, and Iran which together account for more than 50% of the whole region’s population. Hence, it is justified to divide the region into categories. Based on the World Bank classification, the countries of the region could be divided into four income groups according to the income level: low-income group, lower-middle-income group, upper-middle-income group, and high-income group [25].

Health disparities have been documented all across the world. Nutrition is one of the intermediary elements between SES and health outcomes. According to a growing body of evidence, SES has been found to be inversely associated with higher BMI and obesity in high-income countries [26,27]. For middle-income countries, the association is more mixed as some reviews found a positive correlation [28,29] while another review found a negative correlation; however, the latter reported a remarkable positive association between SES and obesity in low-income countries [29,30].

It has been researched that low SES groups in Asia typically consume more carbohydrates and less protein and fat than high SES ones, indicating that SES influences nutrient intake in both developing and developed Asian countries. The low cost of high-energy-dense meals, food price increase, and dietary expertise may all play a role in the link between low SES and obesity [28]. In the EMR, however, the situation is flipped, since processed, high-energy-dense foods are extensively consumed in the high-income EMR countries [31,32]. For children under five, low SES and low maternal education level were significantly associated with a lower prevalence of overweight/obesity compared with high levels of SES and maternal education in Colombia. In contrast, low SES and low mother educational level categories had high prevalence rates of wasting, stunting, and anaemia [33].

Therefore, this analytical review was conducted to clarify the nutrition status among children under five, children and adolescences from 5–19 years, adults and women at reproductive age in the different EMR income groups according to the latest available estimates. This is to investigate the correlations of these nutrition indicators with the income level, as well as to discuss the potential causes of the nutrition problems and describe the current challenges and progress regarding nutrition policies and programmes that could influence the life and well-being of EMR population.

## 2. Methods

In this article, the nutrition status of different income groups in the EMR is discussed and illustrated. Data for the demographics, children under five malnutrition, infant and young child feeding (IYCF), anaemia in women of reproductive age, overweight and obesity among children and adolescents aged 5–19 and adults, in addition to existing nutrition policies and strategies in the EMR are summarized. For overweight and obesity among children and adolescents (aged 5–19 years) as well as adults, age-standardized estimates were obtained from the NCD Risk Collaboration, which in turn, are based on data provided to WHO and the NCD Risk Factor Collaboration [2]. Adjustments were performed by the NCD Risk Factor Collaboration to standardize risk factor definition, age groups, reporting year, and population representativeness for these estimations. To account for disparities in population age/sex composition and to allow comparisons between nations, age-standardized prevalence figures were produced by the NCD Risk Factor Collaboration [2]. Overweight and obesity for adults are defined as BMI of 25 kg/m^2^ or higher and BMI of 30 kg/m^2^ or higher, respectively, while for children and adolescents (aged 5–19 years) they are defined as BMI greater than one and two standard deviations (SD) above the median, respectively. Additional data were collected from recent national surveys [34,35,36,37,38,39,40].

Data regarding the demographics and gross domestic product (GDP) per capita (current US$) were collected from the World Bank [23] and from a national health survey for Afghanistan [41]. Child malnutrition data (wasting, stunting, and overweight prevalence among children under five years) were obtained from the WHO-GHO [20]. For wasting, due to the scarcity of data in GHO, data were also collected from national nutrition surveys [42,43,44,45,46,47,48] and from Eastern Mediterranean Health Observatory [49]. Data regarding IYCF (early initiation of breastfeeding and exclusive breastfeeding) were obtained from UNICEF’s Breastfeeding database [50], Eastern Mediterranean Health Observatory [49], and published national surveys [34,47,51,52]. Data on anaemia among women of reproductive age were obtained from the GHO [20]. Wasting was defined as weight-for-height more than 2 SD below the median and stunting as height-for-age more than 2 SD below the median, while cut-offs for anaemia were haemoglobin concentration less than 120 g/L for non-pregnant women and lactating women, and less than 110 g/L for pregnant women, adjusted for altitude and smoking [20].

The policies related to actions to reduce NCDs in the EMR as well as the polices on healthy diets in the countries of the EMR are tabulated. Data have been extracted from various sources. These include WHO’s global [20] and regional health observatories [49], data collected for the second WHO Global Nutrition Policy Review 2016–2017 [53], the WHO Global database on the Implementation of Nutrition Action (GINA) [54], communication about country-level action from WHO country offices and national government nutrition focal points, and relevant academic papers [55,56,57,58,59,60,61,62,63,64,65,66,67]. Data were collected specifically on the policy areas featured in the WHO Strategy on nutrition for the EMR 2020–2030 [68].

The World Bank classification 2020–2021—which is based on gross national income per capita in current USD—was used to group the EMR countries into different income groups according to the income level. The low-income group included Afghanistan, Somalia, Sudan, Syria, and Yemen. The lower-middle-income group included Djibouti, Egypt, Iran, Morocco, Pakistan, Palestine, and Tunisia. The upper-middle-income group included Iraq, Jordan, Lebanon, and Libya. The high-income group included Bahrain, Kuwait, Oman, Qatar, Saudi Arabia, and United Arab Emirates (UAE) [25].

The regional income groups’ estimates over time for stunting and overweight in children under five, anaemia in women of reproductive age, and overweight and obesity among children and adolescents aged 5–19, as well as adults were calculated by using the weighted average formula [69]. The weight of each country for each year was first calculated by dividing the country’s population with the income group’s total population, after which weighted prevalences were obtained by multiplying the weight with the actual prevalence and finally summing up the weighted prevalences of the income group for each year. There were no data available on child malnutrition for UAE and on anaemia in women of reproductive age for Palestine, hence the total populations of these countries were excluded from the respective calculations. For the indicators wasting, exclusive breastfeeding, and early initiation of breastfeeding, the latest available country-specific estimates were visualized and compared. For wasting, the averages for the income groups for the periods of 2000–2010 and 2011–2021 were also produced by first calculating the average prevalences for the periods 2000–2010 and 2011–2021 for each country using the available estimates, after which the weighted prevalences were obtained with the method described above (except using the calculated average prevalences instead of actual prevalences) and then summed up. All the indicators included in this paper are illustrated in Figure 2.

## 3. Results

### 3.1. Demographics

Reviewing the records on the percentage of 0–14 aged population of total population in EMR income groups revealed that the low-income EMR group had the highest percentage of this age group as compared to other EMR income groups. All EMR low-income countries had a proportion of 0–14 aged population more than 30%, while Afghanistan and Somalia had more than 40%. On the contrary, all high-income countries reported a percentage equal or less than 25% from this age group, Qatar and UAE reported a percentage equal or less than 15% of 0–14 age population. The highest under-five mortality rate (per 1000 live births) was recorded in a low-income country Somalia (117), followed by Pakistan (67) (a lower-middle-income country). The under-five mortality rates (per 1000 live births) among high-income countries were less than 10 except for Oman which reported 11. GDP per capita (current US$), which is commonly used as an indicator for development and economic growth, predictably revealed the highest levels among high-income EMR countries. A high-income country Qatar, with a population size less than 0.4% of the total population of the EMR region, had the highest GDP per capita (50 805.5) while Somalia, a low-income country, had the lowest GDP per capita among EMR countries (309.4) (Table 1).

### 3.2. Child Malnutrition (Children under Five Years)

Throughout the past two decades (2000–2020), under-five stunting was inversely associated with the income level in the EMR, as it was most prevalent in the low-income group, followed by lower- and upper-middle-income groups, whereas high-income group had the lowest prevalence. When comparing the latest country-specific prevalence estimates for wasting in children under five, the highest prevalence rates could be found inside the low-income group, with Sudan having the highest prevalence (14.1% in 2018) (Figure 3, Table 2 and Table 3). The average prevalence calculations for wasting in the periods of 2000–2010 and 2011–2021 show that the deepest decline in wasting prevalence was experienced in the high-income EMR group (Figure 3). All the available country survey estimates from 2000 to 2021 for wasting that were used for the average wasting prevalence calculations are presented in Table 2. For under-five overweight, the upper-middle-income group had the highest prevalence rate throughout the 2000s (Figure 3, Table 3).

A closer look on the data of the high-income group reveals that the prevalence of stunting in children under five in the high-income group decreased from 12.6% to 5.0% over the past two decades (Figure 3, Table 3). For wasting in children under five, the average prevalence decreased from 10.6% in the first decade of 2000s to 4.6% in the second decade of 2000s (Figure 3). Conversely, the prevalence of overweight in children under five in high-income countries increased dramatically from 3.5% to 7.6% between 2000 and 2020 (Figure 3, Table 3).

In the upper-middle-income group, the prevalence of stunting decreased from 23.7% to 14.2% over the past two decades (Figure 3, Table 3). The average prevalence of wasting decreased slightly: the average prevalence was 5.7% in the period of 2000–2010 and 4.6% in the period of 2011–2020 (Figure 3). On the contrary, the prevalence of overweight in children under five in the upper-middle-income group increased from 10.3% to 11.6% between the years 2000 and 2020 (Figure 3, Table 3).

The child malnutrition trajectories in the lower-middle-income group followed similar trends than in the high-income and upper-middle-income groups. The prevalence of stunting decreased from 30.4% to 25.1% during the past two decades (Figure 3, Table 3). The average prevalence of wasting decreased from 9.8% in the period of 2000–2010 to 7.6% in the period of 2011–2020 (Figure 3). The prevalence of under-five overweight increased from 7.9% in 2000 to 8.7% in 2020 (Figure 3, Table 3).

In the low-income group, the prevalence of stunting in children under five decreased from 43.6% to 33.6% over the past two decades (Figure 3, Table 3). The average prevalence of wasting decreased from 12.2% in the first decade of 2000s to 10.5% in the second decade of 2000s (Figure 3). Unlike in the other income groups, the trend for overweight in children under five in the low-income group was diminishing as the prevalence decreased from 6.7% to 4.9% between the years 2000 and 2020 (Figure 3, Table 3).

### 3.3. Infant and Young Child Feeding

Comparison of the latest country-specific estimates shows no correlation between the income level and the prevalence of early initiation of breastfeeding or exclusive breastfeeding. The highest estimate for early initiation of breastfeeding was recorded in Oman (82.0% in 2017), followed by Iran (68.7% in 2010) and Sudan (68.7% in 2014), then Jordan (67.0% in 2017). Highest estimate for exclusive breastfeeding was recorded in Sudan (61.5% in 2018), followed by UAE (59.7% in 2018), and Afghanistan (57.5% in 2018). The lowest estimates, in turn, were recorded in Pakistan (19.6% in 2018), Egypt (27.1% in 2014), and Libya (28.8% in 2014) for early initiation of breastfeeding, and in Kuwait (7.8% in 2019), Yemen (9.7% in 2013), and Djibouti (12.4% in 2012) for exclusive breastfeeding (Table 4 and Figure 4).

### 3.4. Anaemia in Women of Reproductive Age

Throughout the past two decades (2000–2019), anaemia in women of reproductive age—pregnant and non-pregnant women combined—was inversely associated with the income level, as it was most prevalent in the low-income group, followed by the lower- and upper-middle-income groups, whereas high-income group had the lowest prevalence. Interestingly, all the income groups except lower-middle-income group showed a decreasing trend until the years 2012–2014, after which the trend started to increase. The lower-middle-income group showed a relatively stable, though modest, decreasing trend throughout the measuring period (from 36% in 2000 to 34.0% in 2019). The anaemia situation was particularly alarming in the low-income group where the prevalence was over 40%—and therefore could be considered as a severe public health problem—throughout the measuring period (2000–2019) (Table 5, Figure 5).

### 3.5. Overweight in Children and Adolescents Aged 5–19 and in Adults

Throughout the measuring period (2000–2016), overweight prevalence among children and adolescents aged 5–19 years was directly associated with the income level, as it was most prevalent in the high-income group, followed by the upper- and lower-middle-income groups, whereas low-income group had the lowest prevalence throughout the measuring period. Similarly, when comparing the regional income group trajectories for overweight prevalence among adults between the years 2000 and 2016, the highest prevalence rates throughout the measuring period were found in the high-income and upper-middle-income groups while the lower-middle-income and low-income groups revealed the lowest estimates, despite the significant increase in overweight prevalence in all groups.

In high-income and upper-middle-income groups, the prevalence of overweight among children and adolescents aged 5–19 increased from 26.0% to 36.0% and from 22.9% to 32.0%, respectively, between the years 2000 and 2016 (Table 6, Figure 6). In 2016, all high- and upper-middle-income EMR countries were above the global overweight prevalence of 18.4% among children and adolescents [20]. Similarly to children and adolescents, an overall increase in the prevalence of overweight among adults was estimated as the prevalence increased from 60.3% to 69.1% in the high-income group and from 56.4% to 65.4% in the upper-middle-income group between the years 2000 and 2016 (Table 7, Figure 6). In 2016, all high-income EMR countries were above the global rate of overweight among adults (38.9%) [20].

In the lower-middle and low-income groups, the prevalence of overweight among children and adolescents aged 5–19 rose dramatically from 12% to 20.5% and from 8.1% to 15.3% between the years 2000 and 2016 (Table 6, Figure 6) with increase rates of 70.8% and 88.9%, respectively. In 2016, all low-income EMR countries were below the global rate of overweight among children and adolescents (18.4%) except Syria (28.3%) and Yemen (20%) [20]. In parallel, an overall increase in the prevalence of overweight among adults has been estimated in these two groups: from 36.2% to 46.0% in lower-middle-income group and from 28.2% to 35.6% in low-income group between the years 2000 and 2016 (Table 7 and Figure 6). In 2016, all low-income EMR countries were below the global rate of overweight among adults (38.9%) except Syria (61.4%) and Yemen (48.8%) [20].

The latest available data from national surveys revealed that the prevalence rates of overweight—defined as BMI ≥ 25–29.9—among adults in Bahrain (2018) [36], Kuwait (2019) [34], and Lebanon (2017) [40] were 35.5%, 36.4%, and 38%, respectively. Additionally, data from the same Kuwait Nutrition Surveillance System’s Annual report 2019 showed that almost half of the school-aged children were either obese or overweight. The prevalence of overweight in children and adolescents was 20.2% (defined as BMI between 1 and 1.9 SD above the median), while the prevalence of obesity was 28.4% (measured as BMI > +2 SD above the median) [34]. In the same context, data from the STEPwise surveys conducted in Jordan (2019) [37], Morocco (2017) [38], Palestine (2010–2011) [39], and Kuwait (2020) [35] reported the overweight prevalence rates among adults—defined as BMI ≥ 25—were 60.7%, 53%, 57.8% and 74.6%, respectively.

### 3.6. Obesity in Children and Adolescents Aged 5–19 and in Adults

Throughout the observation period (2000–2016), obesity prevalence among children and adolescents aged 5–19 years was directly associated with the income level as it was most prevalent in the high-income group, followed by the upper- and lower-middle-income groups, whereas low-income group had the lowest prevalence. Similarly, when comparing the estimates for obesity prevalence among adults during the same period, the highest prevalence rates could be found among the high and upper-middle-income groups while lower-middle and low-income ones had the lowest estimates, despite the significant increase in obesity prevalence in all income groups.

In the high- and upper-middle-income EMR groups, an overall increase of 70.2% and 72%, respectively, was observed in the prevalence of obesity among children and adolescents aged 5–19 between the years 2000 and 2016. During this time-period, the proportion of obese children and adolescents in the high- and upper-middle-income groups in the EMR increased from 10.4% to 17.7% and from 8.2% to 14.1%, respectively (Table 6 and Figure 6). In 2016, all high-income EMR countries were above double the global rate of obesity among children and adolescents (6.8%) [20]. An overall increase in the prevalence of obesity among adults in these two groups has been recorded as well: from 25.3% to 34.1% in the high-income group and from 23.2% to 31.6% in the upper-middle-income group between the years 2000 and 2016 (Table 7 and Figure 6). In 2016, all high-income EMR countries were more than double above the global rate of obesity among adults (13.1%) [20].

Even more dramatic increase in the prevalence of obesity among children and adolescents aged 5–19 was reported in the lower-middle and low-income EMR groups, demonstrated by the drastic increase rates of 124.3% (from 3.7% to 8.3%) in the lower-middle-income group and 172.2% (from 1.8% to 4.9%) in the low-income group between the years 2000 and 2016 (Table 6 and Figure 6). In 2016, all low-income EMR countries except Syria and Yemen were below the global rate of obesity among children and adolescents (6.8%) [20]. An overall increase in the prevalence of obesity in these two groups during the same period has been witnessed among adults as well: from 12.1% to 19% in the lower-middle-income group and from 7.7% to 12% in the low-income group, by increase rates of 57% and 55.8%, respectively (Table 7 and Figure 6). In 2016, all low- and lower-middle-income EMR countries were yet below the global rate of obesity among adults (13.1%) except Syria and Pakistan [20].

Data from the national health surveys conducted in Bahrain in 2018 [36], and in Kuwait in 2019 and 2020 [34,35] revealed that the obesity prevalence rate (BMI ≥ 30) among adults was 36.9%, 41.7%, and 36.6%, respectively. Additionally, data from the STEPwise surveys conducted in Lebanon 2017 [40], Jordan 2019 [37], Morocco 2017 [38], and Palestine 2010–2011 [39] revealed the prevalence of obesity among adults as 27%, 32.1%, 20%, and 26.8%, respectively.

## 4. Discussion

### 4.1. Findings and Dietary Trends in Relation to the Socioeconomic Status

Many factors interact together to influence the dietary patterns and diet composition in a certain population or community. These factors include economic factors (income and prices), individual preferences and beliefs, cultural traditions, as well as geographical and environmental factors [70]. Although food consumption patterns have generally become more diversified during the past decades worldwide [71], the dietary patterns in the EMR—among other regions—are becoming higher in salt, sugar, and saturated or trans fats, and this change is accompanied by a general decrease in physical activity. This shift, known as a nutrition transition [68], and its consequences are witnessed in the results of this article, especially in the rapidly growing overweight and obesity rates in all income groups.

In this article, countries in the EMR were categorized into four income groups based on socioeconomic status. According to the results, the high-income group could be characterized by high and growing rates of overweight and obesity, low and decreasing rates of stunting and wasting, varying percentages of early initiation and exclusive breastfeeding, and relatively optimal micronutrient status. The trend of overweight and obesity among adults and children and adolescents aged 5–19 was highest in the high-income group out of all EMR income groups throughout the measuring period. Interestingly though, the prevalence of overweight among children under five was lowest—although sharply rising—in the high-income group out of all EMR income groups from 2000 up till 2011, after which the prevalence kept increasing, and the income group stayed third in the ranking of under-five children’s overweight prevalence. The future will show if this indicates even higher rates of overweight and obesity among the older age groups in the high-income region during the following decades when the younger generations grow up.

Already in 1998, Musaiger stated in his article [31] that the Gulf Cooperation Council (GCC) countries, which constitute the high-income EMR income group in this article, had experienced a rapid and drastic change in socioeconomic status, food consumption patterns and lifestyle over the past three decades. The traditional diet, which consisted of items such as dates, milk, rice, brown bread, fish, and vegetables, had been increasingly replaced with white bread, fast foods, and cereals. Additionally, the intake of sugar and fat was on rise. Although recent data in the GCC countries on food consumption patterns are scarce, the available studies reveal that this trend towards westernized diets still holds true. As an example, nationally representative survey conducted in 2013 showed that in Saudi Arabia—with a population of around 70% of the high-income group’s population—white bread was preferred over brown bread, full-fat dairy products over low-fat or skimmed ones, and the consumption of fruits and vegetables was low, while the consumption of sugar-sweetened beverages (SSB) was relatively high, especially among young adults [72]. Ng et al. found out in their study conducted in 2011 that around a third of female adults and adolescents in UAE consumed more calories than their estimated energy requirements, while the proportion of people consuming more calories than needed was even higher among girls and boys aged 6–10 years (43.2% and 38.2%, respectively) [73]. In the same study, almost three in five female adults, nearly half of the female adolescents and more than one in three girls aged 6–10 had low physical activity level [73]. Regarding dietary habits of children under five, Almaamary et al. [74] found out that in Oman the proportion of children aged two to five years who consumed sweetened fruit juices or sweetened coffee/tea once a day or more often was 55% and 68%, respectively. Moreover, 55% of the children had French fries/chips daily or more often, while 70% consumed vegetables and 66% consumed fruits only once a day or less often [74]. Suboptimal food consumption patterns and decreased physical activity are likely to explain the high overweight and obesity trends in the high-income group.

The high-income group is not the only EMR income group with high prevalence rates of overweight and obesity: our results showed that the upper-middle-income group followed the high-income group in overweight and obesity prevalences with about a five-year difference. The prevalence of overweight among children under five years, in turn, was highest in the upper-middle-income group out of all EMR income groups throughout the measuring period (2000–2020). Stunting and anaemia trends in this income group were decreasing and stayed lower than EMR average all time, but the prevalence rates are still worrying. Wasting, on the other hand, was lowest in the upper-middle-income group out of all EMR income groups in 2000s, and as low as the average prevalence in the high-income region in 2010s. Exclusive breastfeeding and early initiation of breastfeeding rates were relatively low.

Among the most alarming notions regarding the upper-middle-income group are the high prevalence rates of overweight and obesity, especially among children under five years. Major contributors to this include increased dietary intake of fats and refined sugar, as well as insufficient fruit and vegetable consumption combined with a sedentary lifestyle with minimal physical activity, all occurring in an environment of aggressive commercial marketing of fast foods and breast-milk substitutes, and the persistence of poverty pockets [75]. In Lebanon for example, two- to five-years-old children were documented to have high intakes of fat, saturated fat, and sugar (38.8%, 12.7%, and 20.3% of energy intake, respectively) in 2012 [76]. Likewise, in Jordan, a low protein intake (3.7% of energy intake), coupled with a high intake of saturated fat (13.7%-14.2% of energy intake) was reported for the same age group [77]. Food consumption patterns of Jordanian adolescents were studied in 2014 by Tayyem et al. [78], whose results revealed that most teenagers did not eat fruits, vegetables, or breakfast daily, while the majority consumed SSB, sweets/chocolates, and French fries/potato chips and at least three times a week. In a study by Nasreddine et al. [79] with Lebanese 4–13 years old children, the majority of the children did not consume recommended amounts of foods from food groups such as dairy, vegetables, fruits, and whole grains, while the intake of added sugars was high, mainly due to high consumption of sweets, sweetened beverages, and desserts. These findings are among the factors for the high overweight rates of the upper-middle-income group.

Lower-middle-income EMR countries constitute the third category in this article with considerable undernutrition, modestly decreasing and close to EMR average rate of anaemia, rising and close to EMR average rates of overweight and obesity, and relatively low breastfeeding rates. The main causes of suboptimal nutrition status in this group are similar to those of the upper-middle-income group, but also weakness of economic infrastructure, lack of consumer education and protection laws, and widespread poverty have a role to play in some of the countries in this income group [75].

It is noteworthy that the results which were calculated by using the weighted average formula (stunting, overweight, obesity, and anaemia) in the lower-middle-income group are strongly determined by the dietary status of Pakistan, Egypt, and Iran—the three most populous countries in the EMR which together account for nearly 90% of the lower-middle-income group’s population. Haider and Zaidi [80] stated in their article from 2016 that the food consumption patterns of Pakistanis on average were lacking diversity and micronutrients, while consumption of high-energy dense food items was high, yet energy intake remained relatively low. This highlights the spectrum of the nutrition problems inside the country, where big differences in nutrition status and food consumption patterns are observed among different subgroups. According to another study, grains, dairy, sweets and fats accounted for almost 70% of all food expenditures [81]. About half of all daily calories came from wheat, which was the main source [81,82]. These notions further emphasize the problem regarding the lack of diversity in the food consumption patterns in Pakistan and in the lower-middle-income group in general, potentially explaining the high prevalence rates of anaemia. Alarming food consumption patterns have been observed in Egypt as well, where one in three adolescents consumed at least three sugary drinks per day [83]. The amount of sugar available for consumption per capita was found to be high in both Egypt and Pakistan (200 g and 150 g/person/day, respectively) [84].

Afghanistan, Somalia, Sudan, Syria, and Yemen fall into the fourth category, called the low-income group, which is experiencing humanitarian crises and facing severe undernutrition, widespread anaemia, and modest but rising rates of overweight and obesity among adults, children and adolescents aged 5–19 years. It is, though, the only EMR income group where the trend of overweight among children under five was decreasing between the years 2000 and 2020. Regarding breastfeeding practices, the results were mixed.

Disruptions in national development programmes and essential nutrition services—including food safety and food control systems, nutrition counselling, and supplementation programmes—are among the potential explanatory factors for the burden of nutritional problems in the low-income countries [75,85]. Furthermore, other main contributors to malnutrition most likely include a lack of access to nutritious and safe foods and affordable healthy diets, inadequate access to sanitation and safe drinking water, exposure to environmental contaminants and poor hygiene practices [85]. Data from 2021 revealed that more than one in five people in Somalia and Sudan were acutely food insecure and in need of urgent assistance (Crisis or worse, IPC/CH Phase 3 or above), while the percentage was as high as over 50% in Afghanistan, Syria and Yemen [86]. The most recent estimates reveal that in some of the countries, such as Somalia, the situation has further worsened during 2022 [87].

According to the Somalia nutrition strategy, the calories of Somalis were derived mainly from staples (46%), oil (14%), and sugar (19%), while nutrient-rich foods such as meat, milk, fruit, and pulses each accounted only for 3–5% of the total calories [88]. Lack of diversity in diet was also observed in Afghanistan, where almost two-thirds of the calories were obtained from wheat, and among the poorest 20% of the households, wheat accounted for three-fourths of the calories in 2003 [89]. The contributing factors for malnutrition among people in Afghanistan include poverty, inadequate education, childbearing in teenage years, gender inequality, insufficient child feeding and caring practices, lack of access to important sanitation facilities as well as inadequate potable water and hygiene [90,91]. It is likely that similar factors are behind the high malnutrition rates of the other countries of the low-income group as well.

To understand the nutrition transition and the results regarding the trends of the nutrition indicators in the low- and middle-income countries of the EMR, it is important to analyse the changes that have occurred in food availability and food consumption [92]. Data from the FAO food balance sheets and from food consumption surveys between 1969–1971 and 2011 highlight a shift towards an increasingly energy-dense diet and high intakes of fat and sugar, coupled with a parallel decrease in the consumption of carbohydrates [93,94,95,96,97,98]. Data from FAO also show that sugar availability, which is reported to be a predisposing factor to obesity, increased considerably during the same period (1969–2011) in several low- and lower-middle-income countries in the EMR, such as Afghanistan, Sudan, Tunisia, as well as Yemen, and doubled in Egypt. However, a reduction was noted in Pakistan [93].

According to our results, the EMR income groups are experiencing both high rates of stunting and overweight among children younger than five years, accompanied with inadequate breastfeeding practices. WHO and UNICEF recommend that breastfeeding is started within one hour of birth (early initiation of breastfeeding) and that babies are exclusively breastfed for the first 6 months of their life (exclusive breastfeeding) [99]. It has been shown that inadequate breastfeeding practices, including very early introduction of complementary foods, increase the risk of obesity and noncommunicable diseases disease later in life [100,101]. The available data highlight the low rates of exclusive breastfeeding in the first 6 months in most countries in the EMR (Table 4, Figure 2). The rates range from 7.8% to 61.5%, hence most of the EMR countries fall below the World Health Assembly’s global nutrition target of 50% by 2025/2030 [102] and the regional target of 70% by 2030 [68], except Sudan (61.5%), UAE (59.7%), and Afghanistan (57.7%) [34,47,49,50,51,52].

The observed correlations regarding the income level and nutrition indicators of this paper are mostly consistent with the ones at global level [20].

### 4.2. Progress and Challenges

#### 4.2.1. Progress in the Development and Implementation of Policies and Programmes

By endorsing the WHO Strategy on nutrition for the EMR (2020–2030) in October 2019, the EMR countries committed to strengthened action on nutrition to achieve food security, end all forms of malnutrition, and improve nutrition throughout the life-course by 2030. By August 2022, all EMR countries except Libya had developed a national nutrition strategy or action plan, and all EMR countries except Afghanistan, Djibouti, Iraq, Pakistan, Somalia, and UAE had a plan for obesity prevention [53,54,55] (Table 8). These strategies aim to enhance the nutritional status by encouraging countries to reframe nutrition as a core of their development agenda. Furthermore, these strategies provide a conceptual framework to help countries choose appropriate nutrition actions needed according to the most critical health problems and their national situation as well as available resources [103].

#### 4.2.2. Code of Marketing of Breast-Milk Substitutes

National legislation should include implementation and continuous monitoring of the International Code of Marketing of Breast Milk Substitutes and related World Health Assembly resolutions [104]. Eighteen countries in the region have issued legislation to protect breastfeeding or implemented a code of marketing of breast milk substitutes with full or partial legal documents [61]. According to the WHO reports on the implementation of the International Code of Marketing of Breastmilk Substitutes, eight countries (Afghanistan, Bahrain, Kuwait, Lebanon, Pakistan, Yemen, Saudi Arabia and UAE) have comprehensive legislation or other legal measures reflecting all or most provisions of the code [54,61,66]. Three countries (Djibouti, Syria and Tunisia) have legal measures containing many provisions of the Code, and seven countries (Jordan, Iran, Iraq, Palestine, Oman, Sudan, and Egypt) have legal measures that contain some provisions of the code [54,61,66]. Three countries are currently studying the issue (Libya, Morocco, and Qatar), and one country (Somalia) has no legal measures [61,66] (Table 8).

**Table 8 children-10-00236-t008:** The implementation of key nutrition policies and strategies in the EMR countries [20,49,53,54,55,56,57,58,59,60,61,62,63,64,65,66,67,68,105,106].

Key National Nutrition Programs (Policies and Strategies)	Countries That Have Implemented the Program/Policy
**National Nutrition Strategy or action plan**	All EMR countries except Libya
**Plan for obesity prevention**	All EMR countries except Afghanistan, Djibouti, Iraq, Pakistan, Somalia and UAE
**Code of marketing of breast milk substitutes**	All EMR countries except Libya, Morocco, Qatar and Somalia
**Food-based dietary guidelines**	All high-income countries Upper-middle-income EMR countries: Jordan, Lebanon and Libya Lower-middle-income EMR countries: Iran, Morocco and Palestine Low-income EMR countries: Afghanistan, Sudan and Syria
**Policy to limit trans fatty acids intake**	All high-income EMR countries All upper-middle-income EMR countries except Libya All lower-middle-income EMR countries except Djibouti
**Policy to reduce salt/sodium consumption**	All high-income EMR countries All upper-middle-income EMR countries except Lebanon and Libya All lower-middle-income EMR countries except Djibouti and Pakistan
**Sugar-sweetened beverages tax**	All high-income EMR countries Lower-middle-income EMR countries: Egypt, Iran, Morocco, Palestine and Tunisia
**Policy to reduce the impact of marketing of food to children**	All high-income EMR countries Upper-middle-income EMR countries: Jordan and Lebanon Lower-middle-income EMR countries: Egypt, Iran, Morocco and Pakistan
**Front-of-pack labelling**	High-income EMR countries: Saudi Arabia and UAE Lower-middle-income EMR countries: Iran, Morocco and Tunisia
**Salt iodization**	All EMR countries
**Flour fortification**	All EMR countries except Libya, Tunisia and Somalia

#### 4.2.3. Policies for Healthy Diet

Most of the EMR countries have developed policies to support healthy diet and to reduce nutrition-related risk factors of NCDs. Fifteen countries in the EMR have some policy to limit trans-fatty acids intake, including all high-income countries, and all upper- and lower-middle-income countries except Djibouti and Libya [20,54,57,105]. Thirteen countries, including all high-income countries and two upper-middle-income countries (Iraq and Jordan) and five lower-middle-income countries (Egypt, Iran, Morocco, Palestine and Tunisia) have a policy to reduce salt/sodium consumption. Djibouti, Lebanon, Libya, and Pakistan and none of the low-income countries have not yet developed such policies [20,54,57] (Table 8).

Twelve countries have developed a policy to reduce the impact of marketing of food to children, including all high-income countries and two upper-middle-income countries (Jordan and Lebanon), and four lower-middle-income countries (Egypt, Iran, Morocco, and Pakistan) [20,54,62,105]. Taxes on SSB have been implemented in 11 countries: all six high-income countries and five lower-middle-income countries (Egypt, Iran, Morocco, Palestine, and Tunisia) [20,55,63,105]. A 50% tax on soft drinks and 100% tax on energy drinks were adopted in GCC countries in 2016 [55]. In 2017, Saudi Arabia led the way in the implementation of the tax, which was later followed by the UAE and then Bahrain. Oman and Qatar started the implementation process in 2019 and Kuwait followed in 2020 [63] (Table 8).

Food-based dietary guidelines (FBDGs) are national documents that offer dietary recommendations suited to local customs, cultures, and dietary patterns, as well as provide the general population with indications for healthy nutrition and daily lifestyles associated with a reduced risk of diseases [65]. FBDGs have been developed in fifteen countries in the EMR, including all high-income countries (Bahrain, Oman, Qatar, Saudi Arabia, Kuwait, and UAE), three upper-middle-income countries (Jordan, Lebanon and Libya), three lower-middle-income countries (Iran, Morocco and Palestine), and three low-income countries (Afghanistan, Sudan and Syria) [20,54,64,105] (Table 8). All FBDGs in the EMR include recommendations for the prevention of obesity, and some countries include advice for the prevention of cardiovascular diseases, blood pressure, and diabetes (six countries: Lebanon, Afghanistan, Iran, Oman, Qatar and Jordan), cancer (three countries: Qatar, Oman, and Iran) and two countries have advice on dental caries (Oman and Lebanon). The FBDGs of Afghanistan, Lebanon, Oman, and Jordan include recommendations on the nutritional status of pregnant women. Jordan, Palestine, Afghanistan, Lebanon, Oman, and Qatar include in their FBDGs advice for breastfeeding women. Additionally, some of the FBDGs in the EMR contain recommendations for specific population groups (children, adolescents and older adults). The FBDGs of Lebanon and Qatar have included recommendations for vegetarians, and the Lebanese guidelines include advice for vegans and lactose-intolerant people as well. Jordan, Saudi Arabia, and Palestine FBDGs include recommendations for nutrition labelling, while the FBDGs of Afghanistan, Iran, Lebanon, Oman, and Qatar include recommendations to prevent micronutrient deficiencies [64,65] (Table 8).

Front-of-pack nutrition labelling (FOPL) has been applied in five countries in the EMR, including two high-income countries (Saudi Arabia and the UAE) and three lower-middle-income countries (Iran, Morocco, and Tunisia) [54,55,56]. Implementation of this programme in the EMR remains limited. Three types of FOPLs are in use or under development in the EMR, namely, traffic light labelling systems, Nutri-Score, and health logos [56]. Three countries in the Region have introduced traffic light labelling, including Iran (mandatory), Saudi Arabia (voluntary), and UAE (mandatory) in 2017, 2018, and 2022, respectively [54,55,106]. In 2017, it was reported that 80% of food products in Iran were labelled with the traffic light label [55]. Tunisia is introducing a health tick logo for the healthiest food products [56] and Morocco is on the way for the Nutri-Score logo use [55,59] (Table 8).

#### 4.2.4. Flour Fortification and Micronutrient Supplementation 

Flour fortification has been applied through voluntary and mandatory regulations in all EMR countries, except Libya, Somalia and Tunisia. The most common nutrients that have been used to fortify wheat flour in the EMR are iron and folic acid (15 and 19 countries, respectively), in addition to B vitamins in 9 countries. Currently, fortification with vitamin A and/or zinc (6 countries) and vitamin D (2 countries) is uncommon in the region [60,67] (Table 8). In 1991, the WHO urged all countries to scale up salt iodization [60,67]. Salt iodization has been implemented in all EMR countries through voluntary and mandatory regulations [107,108,109] (Table 8).

#### 4.2.5. Improving Food Composition Data in the Region

Understanding population diets, guiding the development of policies, and monitoring the impact of those policies all depend on having reliable, pertinent and recent data on the composition of regularly consumed foods. Since 2016, the WHO-EMRO in collaboration with its partners have been assisting the Member States with updating and consolidating their databases on food composition and providing technical support to the EMR countries to expand and raise the available data on food composition [55,68]. After a regional project on capacity development in the use of improved standardized methodologies to update food composition data that was organized with technical partners, twelve countries in the region developed and updated their national food composition databases totally or partially according to WHO-FAO recommendations (Bahrain, Egypt, Iran, Jordan, Kuwait, Lebanon, Morocco, Oman, Pakistan, Palestine, Sudan, and Tunisia) [55]. Efforts are being made to expand the database to cover all the EMR countries in the future [105].

### 4.3. Challenges

#### 4.3.1. The Accelerating Overweight and Obesity among All Age Groups 

Our results show that the prevalence of obesity more than doubled in 20 years among children and adolescents aged 5–19 years in low- and lower-middle-income EMR income groups, and so did the prevalence of overweight among children under five years in the high-income group. Furthermore, over 70% increase in the prevalence rate was witnessed in the prevalence of overweight among children and adolescents aged 5–19 years in the low- and lower-middle-income groups, as well as in the prevalence of obesity among the same age group in the upper-middle- and high-income groups. Increases in the prevalence rates of overweight and obesity were observed also among adults in all EMR income groups, where the increase rates varied from 14.6% to 55.8%. Overweight among children under five years in the low-income group was the only overweight/obesity indicator that decreased between the measuring period 2000–2020.

These results indicate an overall increase in overweight and obesity in all age groups throughout the EMR income groups. Given the current evidence linking childhood and adolescent obesity to an increased risk of obesity and morbidity in adulthood [110,111], the dramatically rising rates among children and adolescents are especially concerning. According to WHO, overweight predisposes to health consequences such as cardiovascular disease, type 2 diabetes, musculoskeletal disorders, and some cancers, which in turn can cause premature death and substantial disability [112]. The rapid rise of overweight and obesity in the low- and lower-middle-income groups of the EMR is of particular concern, given that the health care systems in these countries are not always well-functioning and they may not provide equal access for everyone, hence often leaving the most vulnerable people without proper care or pushing them to poverty due to health care payments [113]. The increasing rates of overweight and obesity call for immediate actions to combat the expanding problem. The Regional framework for action on obesity prevention 2019–2023 for the WHO EMR [114] presents several priority areas and interventions for action, including but not limited to regulatory measures, reformulation of foods, health sector interventions, and surveillance.

#### 4.3.2. Persistent Malnutrition Exacerbated by Local and Global Conflicts and Crises

Low-income EMR countries Afghanistan, Somalia, Syria, Sudan, and Yemen have been facing acute and continued crises during the past decades. The impact of protracted conflicts—including disruptions to food production, markets, and livelihoods; widespread displacement; and economic crisis—are the main drivers for food crises. Other contributors include economic shocks and weather extremes, such as flooding and droughts, and recently also the impact of COVID-19 pandemic and the war in Ukraine [86]. Despite the progress made in the low-income EMR income group in terms of reducing child malnutrition and anaemia during the past two decades, these countries continue to witness the highest prevalence rates of stunting and wasting in under five children and anaemia in women of reproductive age, as the results of this paper show. In many countries, the on-going crises have led to an increase in food insecurity, jeopardizing the progress made in reducing malnutrition during the past decades. In Sudan, Syria, and Yemen, around 93 000 children were being treated for severe acute malnutrition at the beginning of the COVID-19 pandemic. In Afghanistan, millions of people have been severely affected by decades of conflict and displacement, combined with chronic poverty, COVID-19 pandemic, severe drought, a failing health system, and an economy on the brink of collapse [115], pushing around half of the population in need of humanitarian assistance [116]. Additionally, in Somalia, food insecurity and malnutrition are aggravated by localized flooding, drought, and conflicts that have disrupted livelihoods and hindered economic progress and development. In June 2022, 7.1 million people in Somalia were food insecure. Moreover, essential nutrition services, such as iron and folic acid supplements and dietary counselling, are nowadays often unavailable to pregnant and breastfeeding women in low-income countries [85,117].

#### 4.3.3. The Impact of COVID-19 Pandemic and Ukraine’s War on Food Security and Nutrition

In addition to local emergencies, the COVID-19 pandemic has placed a heavy toll on the countries that have already been struggling with food crises. COVID-19 restrictions such as border closures and lockdowns, alongside reduced incomes and disrupted social protection programmes, compromised access to food and forced people to rely on less nutritious foods globally and in the EMR [85,86,118]. At global level, moderate or severe food insecurity increased in 2020—the year when the pandemic spread across the world—nearly as much as in the previous five years combined [9]. The impact of COVID-19 pandemic on nutrition indicators is not seen yet in the results of this paper given that the estimates date back to the time before the pandemic, but it is estimated that the pandemic will increase maternal and child undernutrition as well as child mortality globally, especially in low- and middle-income countries. For example, the Standing Together for Nutrition (ST4N) Consortium has estimated that in a moderate scenario, 9.3 million additional children under five years will be wasted in 2020–2022 and 2.6 million additional children will be stunted in 2022 compared to 2019 in 118 low- and middle-income countries [118]. Potentially, the COVID-19 crisis could cause dramatic direct and indirect nutritional consequences that could extend to future generations, affecting the physical and cognitive development of children and therefore jeopardizing the health status, human capital development, and economic growth of the next generation [118]. Future research will reveal the realized consequences of the pandemic on food security and nutrition.

Another major threat to global food security is the ongoing war in Ukraine, which has been obstructing global supplies of wheat, maize, and other crops, putting further pressure on prices, and making it more difficult for many countries to achieve food security [85,86]. The impact is particularly significant in the EMR where many countries import a big share of their staples, especially wheat, from Ukraine and Russia, remaining heavily exposed to fluctuations in prices and supply levels from these countries. Sudan, for example, obtains around 93% of national wheat imports from Russia and Ukraine [86]. Globally, food prices hit an all-time high in 2022 [85,119], demonstrating the effects of global crises on the global economics. The extent of the impact of the Ukraine war on food security globally and in the EMR will depend on the course of the conflict and actions taken. In 82 countries where the WFP has operational presence and data is available, the number of people who are acutely food insecure or at high risk has already surpassed a record high of 345 million people in 2022. This is driven by the consequences of the war in Ukraine [120].

Food insecurity adversely affects the availability of healthy diets, purchasing power, and dietary patterns and has a detrimental impact on the nutritional status especially in children, adolescent girls, and women [85]. Studies conducted in Palestine, Iran, and Bangladesh demonstrated that children under five experiencing food insecurity had higher prevalence of stunting, wasting, thinness and/or underweight compared to food secure children [121,122,123]. In another study from Bangladesh, women experiencing food insecurity were about 1.6 times more likely to suffer from anaemia compared to their food secure counterparts [124]. Childhood malnutrition is a common underlying cause for deaths in children under five, but it also crucially impairs the child’s ability to reach their full potential in life in terms of development, academic performance and productivity [125]. If the food security situation keeps worsening in the countries of the EMR, it would not be surprising to witness a change of direction towards worse in malnutrition indicator trends, including a rise in stunting and wasting, as well as in anaemia and micronutrient deficiencies. Therefore, it is crucial to use all efforts to reduce food insecurity.

### 4.4. Scarcity of Data

Several gaps and challenges have been identified in the existing nutritional and dietary assessment studies in the EMR particularly in the high- and low-income countries. This includes lack of recent studies examining the nutritional status of children younger than five years, scarcity of national studies evaluating micronutrient deficiencies, and scarcity of nationally representative dietary intake studies in children [73]. The paucity of data in low-income countries is mainly due to political turmoil, instability and limited research funding [126]. It is also important to note that dietary assessment in these countries may be limited by the availability of up-to-date and complete food composition tables, specifically for traditional and culture specific foods, highlighting the immediate need for concerted efforts in this area. The importance of up-to-date nutrition data for decision making was underlined in a report by WHO and UNICEF [127] which concluded that data presented in convincing and interpretative ways develop awareness about nutrition issues among key stakeholders and accelerates the knowledge translation into action by engaging policymakers to make sustainable political commitments for improving nutrition. The lack of recent data regarding some countries and some indicators is also the major limitation of this article. Moreover, the different severity levels of anaemia were not addressed in this article although sub classification would give more accurate image about the intensity of the problem and the need for interventions.

### 4.5. Conclusions, Recommendations, and Future Prospective

This analytical review highlights the situation and the progress achieved in the different EMR income groups through elaborating the prevalence rates of nutritional problems among children under five, children, and adolescences from 5–19 years, adults and women at reproductive age as well as discussing the causes and complexities of the current situation. Accordingly, dedicated and determined efforts are needed by policymakers and practitioners in Member States to control the exposure of EMR populations to unhealthy food, malnutrition, food insecurity, obesity, and the risk of diseases. First, efforts need to be directed to better understand the epidemiology of nutritional disorders in countries. Second, Member States need to be supported to translate the knowledge into actions based on their national priorities. This could be achieved through supporting countries in implementing strategies, policies, and action plans to achieve food security, end all forms of malnutrition, prevent overweight and obesity in the region, and improve nutrition throughout the life stages based on WHO recommendations, such as the ones mentioned in the Strategy on nutrition for the EMR 2020–2030 [68,128]. Third, progress and achievements need to be measured and reported as well as gaps and challenges need to be identified on regular basis, guiding the way forward in each country to improve its capability to achieve the expected outcomes and targets [128]. Finally, interventions and a broad array of policies aiming to improve the nutritional status of the population, achieve food security, end all forms of malnutrition, particularly among vulnerable population groups such as children and women, in addition to prevent obesity, overweight, and the risk factors contributing to NCDs, should be given the priority to be implemented in the regional public health agendas.

The data gathered in this review calls for immediate action to address the region’s low rates of early initiation of breastfeeding and exclusive breastfeeding, in order to improve nutrition status among all age groups. This could be attained by implementing the International Code of Marketing of Breast-Milk Substitutes and the Baby-Friendly Hospital Initiative within health systems [68,104,128]. Regarding the overall increase in the prevalence of overweight and obesity in all age groups in the EMR income groups, there is an urgent need to highlight and instigate government-led reformulating programmes for foods and beverages to reduce the intake of fat and sugar from foods and beverages [104,128], and implement the WHO set of recommendations to reduce the impact of marketing of unhealthy foods and non-alcoholic beverages to children, issue and implement taxes on SSB and foods, and develop guidelines for the provision of healthy food in schools, hospitals and other public institutions [68]. 

These priorities should guide policymakers and other stakeholders, including members of the research community, funding agencies, nongovernmental organizations, and the private sector, to conduct relevant research to address knowledge gaps and to develop culture-specific and evidence-based intervention strategies aimed at improving the nutritional status of the population in these countries.

## Figures and Tables

**Figure 1 children-10-00236-f001:**
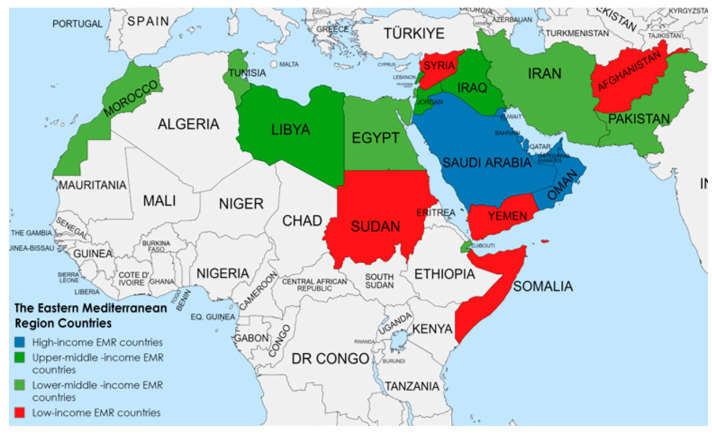
The countries of the Eastern Mediterranean Region.

**Figure 2 children-10-00236-f002:**
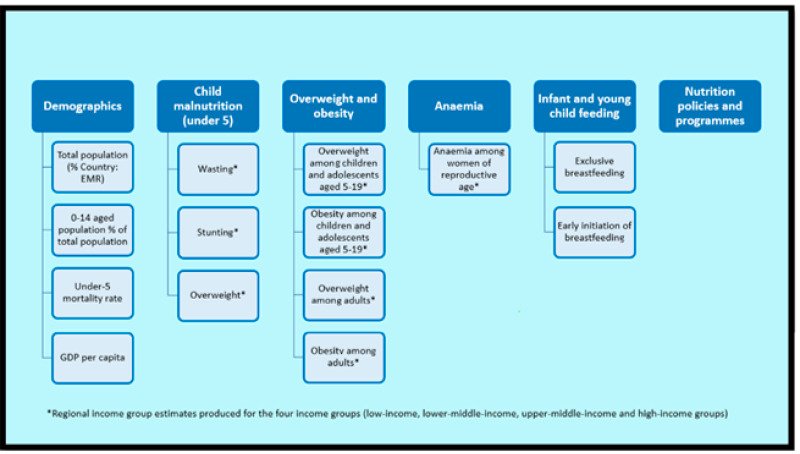
The analyzed indicators.

**Figure 3 children-10-00236-f003:**
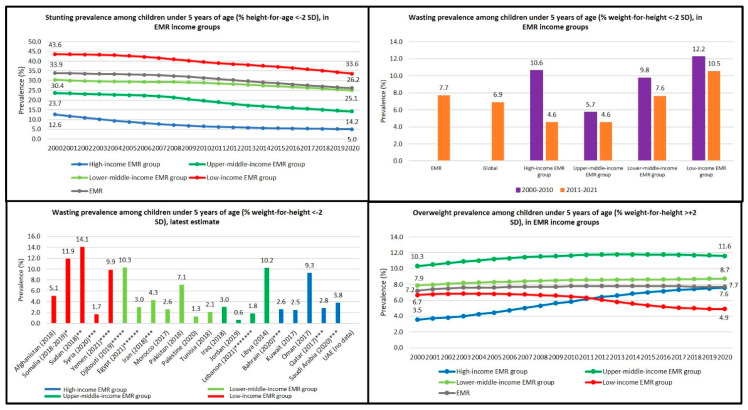
Children under 5 malnutrition trends in the EMR income groups 2000–2020 [20,42,43,44,46,47,48,49]. Sources for the latest country estimates on wasting: * Somalia: [46], ** Sudan: [47], *** Syria, Iran, Bahrain, Qatar, Saudi Arabia: [49], **** Yemen: [48], ***** Djibouti: [44], ****** Egypt: [42], ******* Lebanon: [43], rest of the countries: [20].

**Figure 4 children-10-00236-f004:**
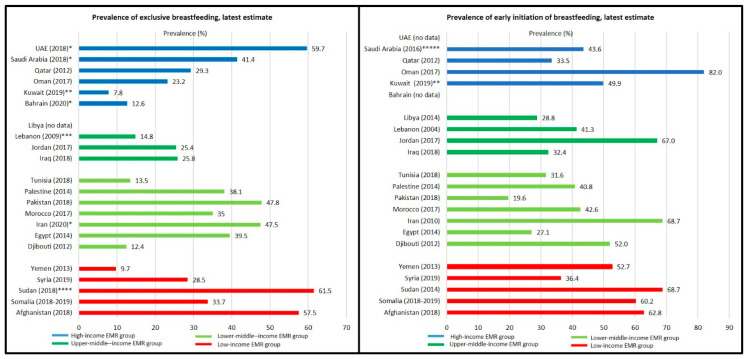
Infant and young child feeding in EMR countries [34,47,49,50,51,52]. Sources for exclusive breastfeeding: * UAE, Saudi Arabia, Bahrain Iran: [49], ** Kuwait: [34], *** Lebanon: [52], **** Sudan [47], rest of the countries: [50]. Sources for early initiation of breastfeeding: ** Kuwait: [34], ***** Saudi Arabia: [51], rest of the countries: [50].

**Figure 5 children-10-00236-f005:**
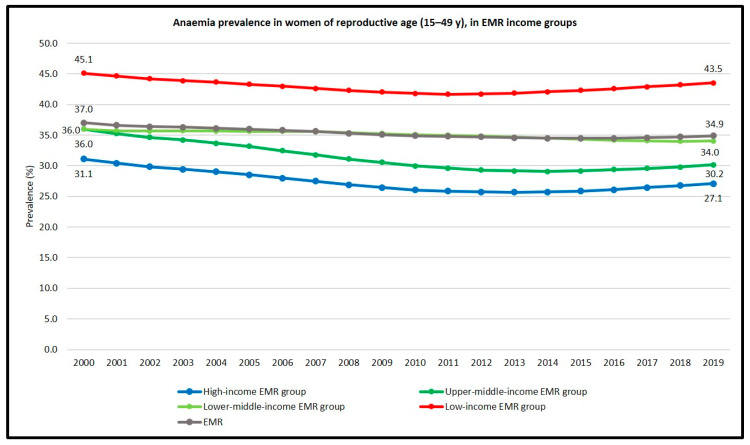
Anaemia prevalence trend in women of reproductive age (15–49 y) in the EMR income groups 2000–2019 [20].

**Figure 6 children-10-00236-f006:**
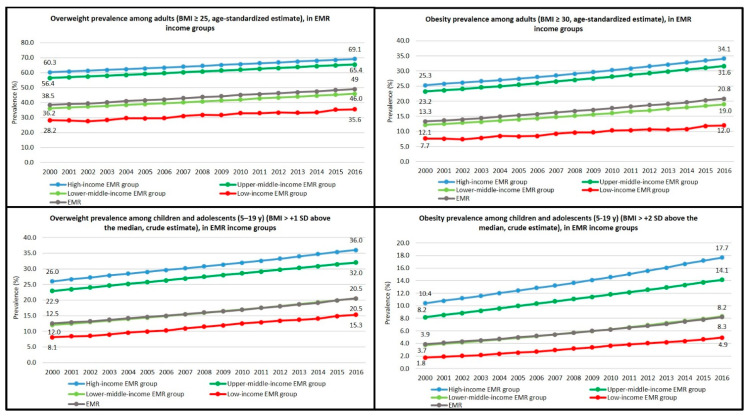
Overweight and obesity prevalence trends among children, adolescents, and adults in the EMR income groups 2000–2016 [20].

**Table 1 children-10-00236-t001:** Demographics of the EMR countries (latest estimates) [23,41].

		Total Population (% Country: EMR) (2020)	0–14 Aged Population % of Total Population (2020)	Under-5 Mortality Rate (per 1000 Live Births)	GDP per Capita (Current US$)
Low-income EMR countries	Afghanistan	38 928 341 (5.3%)	41.8%	50 (2018)	508.8 (2020)
	Somalia	15 893 219 (2.2%)	46.1%	117 (2019)	309.4 (2020)
	Sudan	43 849 269 (6%)	39.8%	58 (2019)	595.5 (2020)
	Syria	17 500 657 (2.4%)	30.8%	22 (2019)	No data
	Yemen	29 825 968 (4.1%)	38.8%	58 (2019)	824.1 (2018)
Lower-middle-income EMR countries	Djibouti	988 002 (0.1%)	28.9%	58 (2019)	3 425.5 (2020)
	Egypt	102 334 403 (14%)	33.9%	20 (2019)	3 547.9 (2020)
	Iran	83 992 953 (11.5%)	24.7%	14 (2019)	2 282.6 (2020)
	Morocco	36 910 558 (5.1%)	26.8%	21 (2019)	3 009.2 (2020)
	Pakistan	220 892 331 (30.2%)	34.8%	67 (2019)	1 193.7 (2020)
	Palestine	4 803 269 (0.7%)	38.4%	19 (2019)	3 239.7 (2020)
	Tunisia	11 818 618 (1.6%)	24.3%	17 (2019)	3 319.8 (2020)
Upper-middle-income EMR countries	Iraq	40 222 503 (5.5%)	37.7%	26 (2019)	4 157.5 (2020)
	Jordan	10 203 140 (1.4%)	32.9%	16 (2019)	4 282.8 (2020)
	Lebanon	6 825 442 (0.9%)	25.1%	7 (2019)	4 891 (2020)
	Libya	6 871 287 (0.9%)	27.8%	12 (2019)	3 699.2 (2020)
High-income EMR countries	Bahrain	1 701 583 (0.2%)	18.3%	7 (2019)	23 443.4 (2019)
	Kuwait	4 270 563 (0.6%)	21.5%	8 (2019)	32 373.3 (2019)
	Oman	5 106 622 (0.7%)	22.5%	11 (2019)	15 343 (2019)
	Qatar	2 881 060 (0.4%)	13.6%	7 (2019)	50 805.5 (2020)
	Saudi Arabia	34 813 867 (4.8%)	24.7%	7 (2019)	20 110.3 (2020)
	UAE	9 890 400 (1.4%)	14.8%	8 (2019)	43 103.3 (2019)

**Table 2 children-10-00236-t002:** Wasting prevalence among children under five years of age in the EMR countries [20,42,43,44,45,46,47,48,49].

		Wasting Prevalence (% Weight-for-Height < −2 SD) among Children under Five Years of Age, Latest Estimate	Wasting Prevalence (% Weight-for-Height < −2 SD) among Children under Five Years of Age, Previous Estimates
Low-income EMR countries	Afghanistan (2018)	5.1	8.6 (2004), 9.5 (2013)
	Somalia (2018–2019)	11.9	13.3 (2006), 14.3 (2009)
	Sudan (2018)	14.1	14.5 (2006), 15.4 (2010), 16.3 (2014)
	Syria (2020)	1.7	4.9 (2000), 10.2 (2001), 10.3 (2006), 11.5 (2010)
	Yemen (2013)	16.4	15.3 (2003), 13.8 (2005), 13.3 (2011)
Lower-middle-income EMR countries	Djibouti (2019)	10.3	19.4 (2002), 10.0 (2010), 21.5 (2012), 17.8 (2013)
	Egypt (2014)	9.5	5.2 (2003), 5.3 (2005), 7.9 (2008), 9.5 (2014)
	Iran (2018)	4.3	4.8 (2004), 4.0 (2010)
	Morocco (2017)	2.6	10.8 (2003), 2.3 (2011), 2.6 (2017)
	Pakistan (2018)	7.1	14.1 (2001), 14.9 (2011), 10.5 (2013), 7.1 (2018)
	Palestine (2020)	1.3	2.0 (2000), 2.5 (2002) 3.2 (2004), 1.8 (2006), 3.3 (2010), 1.2 (2014)
	Tunisia (2018)	2.1	2.9 (2000), 3.4 (2006), 2.8 (2012)
Upper-middle-income EMR countries	Iraq (2018)	3.0	6.6 (2000), 5.6 (2003), 6.9 (2004), 5.8 (2006), 6.5 (2011)
	Jordan (2019)	0.6	2.5 (2002), 1.6 (2009), 2.4 (2012)
	Lebanon (2021)	1.8	6.6 (2004)
	Libya (2014)	10.2	6.5 (2007)
High-income EMR countries	Bahrain (2020)	2.6	2.1 (2014), 3.2 (2017)
	Kuwait (2017)	2.5	2.2 (2001), 2.2 (2002), 2.6 (2003), 3.4 (2004), 3.3 (2005), 2.8 (2006), 3.6 (2007), 2.2. (2008), 1.8 (2009), 2.4 (2010), 1.7 (2011), 2.4 (2012), 2.7 (2013), 2.4 (2014), 3.1 (2015), 2.4 (2016)
	Oman (2017)	9.3	7.1 (2009), 7.5 (2014)
	Qatar (2017)	2.8	10.1 (2013)
	Saudi Arabia (2020)	3.8	11.8 (2004), 4.1 (2017), 4.8 (2018)
	UAE (no data)		

**Table 3 children-10-00236-t003:** Stunting and overweight prevalence among children under five years of age in the EMR income groups [20].

	Stunting Prevalence (% Height-for-Age < −2 SD) among Children under Five Years of Age					Overweight Prevalence (% Weight-for-Height > +2 SD) among Children under Five Years of Age				
	EMR	Low-Income EMR Group	Lower-Middle-Income EMR Group	Upper-Middle-Income EMR Group	High-Income EMR Group	EMR	Low-Income EMR Group	Lower-Middle-Income EMR Group	Upper-Middle-Income EMR Group	High-Income EMR Group
**2000**	33.9	43.6	30.4	23.7	12.6	7.2	6.7	7.9	10.3	3.5
**2001**	33.8	43.5	30.0	23.4	11.7	7.4	6.8	8.0	10.5	3.7
**2002**	33.6	43.4	29.8	23.1	10.9	7.5	6.8	8.1	10.7	3.8
**2003**	33.5	43.3	29.6	23.0	10.1	7.6	6.8	8.2	10.9	4.0
**2004**	33.4	43.1	29.5	22.8	9.4	7.6	6.8	8.2	11.0	4.3
**2005**	33.2	42.7	29.4	22.5	8.8	7.6	6.8	8.3	11.2	4.5
**2006**	33.0	42.2	29.4	22.3	8.2	7.7	6.8	8.3	11.3	4.7
**2007**	32.8	41.6	29.3	22.0	7.7	7.7	6.7	8.4	11.4	5.0
**2008**	32.4	41.0	29.3	21.4	7.2	7.7	6.6	8.4	11.5	5.3
**2009**	32.0	40.3	29.1	20.5	6.8	7.7	6.6	8.5	11.6	5.6
**2010**	31.4	39.6	28.9	19.7	6.5	7.8	6.5	8.6	11.6	5.8
**2011**	30.9	39.0	28.6	18.9	6.2	7.8	6.3	8.6	11.7	6.2
**2012**	30.3	38.5	28.2	18.0	6.0	7.8	6.1	8.6	11.8	6.4
**2013**	29.7	38.1	27.9	17.2	5.8	7.8	5.8	8.6	11.8	6.6
**2014**	29.1	37.6	27.5	16.8	5.6	7.8	5.6	8.6	11.8	6.8
**2015**	28.6	37.1	27.1	16.4	5.5	7.8	5.4	8.6	11.8	7.0
**2016**	28.1	36.6	26.7	15.9	5.4	7.8	5.2	8.6	11.8	7.2
**2017**	27.6	35.8	26.3	15.5	5.3	7.8	5.1	8.7	11.7	7.3
**2018**	27.1	35.1	25.9	15.1	5.2	7.7	5.0	8.7	11.7	7.4
**2019**	26.6	34.4	25.4	14.6	5.2	7.7	4.9	8.7	11.7	7.5
**2020**	26.2	33.6	25.1	14.2	5.0	7.7	4.9	8.7	11.6	7.6

**Table 4 children-10-00236-t004:** Prevalence of exclusive and early initiation of breastfeeding in the EMR countries [34,47,49,50,51,52].

	Prevalence of Exclusive Breastfeeding (Children < 6 Months)		Prevalence of Early Initiation of Breastfeeding (within 1 h of Birth)	
Low-income EMR countries	Afghanistan (2018)	57.5	Afghanistan (2018)	62.8
	Somalia (2018–2019)	33.7	Somalia (2018–2019)	60.2
	Sudan (2018)	61.5	Sudan (2014)	68.7
	Syria (2019)	28.5	Syria (2019)	36.4
	Yemen (2013)	9.7	Yemen (2013)	52.7
Lower-middle-income EMR countries	Djibouti (2012)	12.4	Djibouti (2012)	52.0
	Egypt (2014)	39.5	Egypt (2014)	27.1
	Iran (2020)	47.5	Iran (2010)	68.7
	Morocco (2017)	35	Morocco (2017)	42.6
	Pakistan (2018)	47.8	Pakistan (2018)	19.6
	Palestine (2014)	38.1	Palestine (2014)	40.8
	Tunisia (2018)	13.5	Tunisia (2018)	31.6
Upper-middle-income EMR countries	Iraq (2018)	25.8	Iraq (2018)	32.4
	Jordan (2017)	25.4	Jordan (2017)	67.0
	Lebanon (2009)	14.8	Lebanon (2004)	41.3
	Libya	No data	Libya (2014)	28.8
High-income EMR countries	Bahrain (2020)	12.6	Bahrain	No data
	Kuwait (2019)	7.8	Kuwait (2019)	49.9
	Oman (2017)	23.2	Oman (2017)	82.0
	Qatar (2012)	29.3	Qatar (2012)	33.5
	Saudi Arabia (2018)	41.4	Saudi Arabia (2016)	43.6
	UAE (2018)	59.7	UAE	No data

**Table 5 children-10-00236-t005:** Anaemia prevalence in women of reproductive age (15–49 y) in the EMR income groups [20].

	Anaemia Prevalence in Women of Reproductive Age (15–49 y)				
	EMR	Low-Income EMR Group	Lower-Middle-Income EMR Group	Upper-Middle-Income EMR Group	High-Income EMR Group
**2000**	37.0	45.1	36.0	36.0	31.1
**2001**	36.6	44.6	35.7	35.3	30.4
**2002**	36.4	44.2	35.7	34.6	29.8
**2003**	36.3	43.9	35.7	34.2	29.4
**2004**	36.1	43.6	35.7	33.7	29.0
**2005**	36.0	43.3	35.6	33.2	28.5
**2006**	35.8	43.0	35.6	32.5	28.0
**2007**	35.6	42.6	35.6	31.8	27.5
**2008**	35.3	42.3	35.4	31.1	26.9
**2009**	35.1	42.0	35.2	30.5	26.5
**2010**	34.9	41.8	35.1	30.0	26.1
**2011**	34.8	41.7	35.0	29.6	25.8
**2012**	34.7	41.7	34.9	29.3	25.7
**2013**	34.6	41.8	34.7	29.2	25.7
**2014**	34.5	42.1	34.5	29.1	25.7
**2015**	34.5	42.3	34.3	29.2	25.9
**2016**	34.5	42.6	34.2	29.4	26.1
**2017**	34.6	42.9	34.1	29.6	26.5
**2018**	34.7	43.2	34.0	29.8	26.7
**2019**	34.9	43.5	34.0	30.2	27.1

**Table 6 children-10-00236-t006:** Overweight and obesity prevalence among children and adolescents (5–19 y) in the EMR groups [20].

	Overweight Prevalence (BMI > +1 SD above the Median, Crude Estimate) among Children and Adolescents (5–19 y)					Obesity Prevalence (BMI > +2 SD above the Median, Crude Estimate) among Children and Adolescents (5–19 y)				
	EMR	Low-Income EMR Group	Lower-Middle-Income EMR Group	Upper-Middle-Income EMR Group	High-Income EMR Group	EMR	Low-Income EMR Group	Lower-Middle-Income EMR Group	Upper-Middle-Income EMR Group	High-Income EMR Group
**2000**	12.5	8.1	12.0	22.9	26.0	3.9	1.8	3.7	8.2	10.4
**2001**	12.9	8.4	12.5	23.5	26.6	4.1	1.9	4.0	8.5	10.8
**2002**	13.2	8.5	13.0	24.0	27.2	4.3	2.0	4.2	8.8	11.2
**2003**	13.7	8.9	13.4	24.6	27.9	4.5	2.2	4.4	9.2	11.6
**2004**	14.2	9.6	13.9	25.2	28.4	4.7	2.4	4.7	9.6	12.0
**2005**	14.6	9.9	14.4	25.7	29.0	5.0	2.5	4.9	10.0	12.4
**2006**	15.0	10.3	14.9	26.3	29.6	5.2	2.7	5.1	10.3	12.8
**2007**	15.5	11.0	15.4	26.9	30.2	5.4	2.9	5.4	10.7	13.2
**2008**	16.0	11.5	15.9	27.4	30.8	5.7	3.2	5.7	11.1	13.6
**2009**	16.4	11.9	16.4	28.0	31.3	6.0	3.4	6.0	11.4	14.1
**2010**	16.9	12.5	17.0	28.5	32.0	6.2	3.6	6.2	11.8	14.6
**2011**	17.5	12.9	17.5	29.1	32.6	6.5	3.8	6.6	12.2	15.0
**2012**	18.0	13.4	18.1	29.7	33.3	6.8	4.0	6.9	12.5	15.5
**2013**	18.6	13.7	18.7	30.3	34.0	7.1	4.2	7.2	12.9	16.1
**2014**	19.1	14.1	19.3	30.8	34.7	7.5	4.4	7.6	13.3	16.7
**2015**	19.9	14.9	19.9	31.4	35.4	7.8	4.7	7.9	13.7	17.2
**2016**	20.5	15.3	20.5	32.0	36.0	8.2	4.9	8.3	14.1	17.7

**Table 7 children-10-00236-t007:** Overweight and obesity prevalence among adults in the EMR groups [20].

	Overweight Prevalence (BMI ≥ 25, Age-Standardized Estimate) among Adults					Obesity Prevalence (BMI ≥ 30, Age-Standardized Estimate) among Adults				
	EMR	Low-Income EMR Group	Lower-Middle-Income EMR Group	Upper-Middle-Income EMR Group	High-Income EMR Group	EMR	Low-Income EMR Group	Lower-Middle-Income EMR Group	Upper-Middle-Income EMR Group	High-Income EMR Group
**2000**	38.5	28.2	36.2	56.4	60.3	13.3	7.7	12.1	23.2	25.3
**2001**	39.0	28.1	36.7	57.0	60.8	13.6	7.6	12.5	23.6	25.8
**2002**	39.4	27.5	37.3	57.5	61.3	13.9	7.4	12.8	24.0	26.2
**2003**	40.2	28.4	37.8	58.0	61.8	14.4	7.8	13.2	24.5	26.6
**2004**	41.0	29.7	38.4	58.5	62.3	14.9	8.4	13.6	25.0	27.0
**2005**	41.6	29.5	39.0	59.1	62.9	15.3	8.4	13.9	25.5	27.43
**2006**	42.1	29.6	39.6	59.6	63.4	15.7	8.5	14.3	26.0	28.0
**2007**	43.0	31.1	40.2	60.2	64.0	16.2	9.2	14.7	26.5	28.5
**2008**	43.7	31.8	40.8	60.8	64.5	16.7	9.6	15.1	27.0	29.0
**2009**	44.3	31.7	41.4	61.3	65.1	17.1	9.7	15.6	27.6	29.6
**2010**	45.1	33.0	42.0	62.0	65.7	17.7	10.3	16.0	28.1	30.3
**2011**	45.7	33.0	42.8	62.6	66.2	18.2	10.4	16.6	28.7	30.8
**2012**	46.3	33.4	43.3	63.1	66.8	18.7	10.6	16.9	29.2	31.5
**2013**	46.9	33.2	44.0	63.7	67.4	19.1	10.6	17.5	29.8	32.1
**2014**	47.5	33.4	44.6	64.3	68.0	19.6	10.8	17.9	30.4	32.8
**2015**	48.4	35.3	45.3	64.9	68.5	20.3	11.8	18.4	31.0	33.4
**2016**	49.0	35.6	46.0	65.4	69.1	20.8	12.0	19.0	31.6	34.1

## Data Availability

The new data regarding the calculated weighted average for all EMR income group are represented in the study Table 1, Table 2, Table 3, Table 5, Table 6 and Table 7.

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
