# Peer review of "Nutrition Profile for Countries of the Eastern Mediterranean Region with Different Income Levels: An Analytical Review"

_children, 2023, doi:10.3390/children10020236_

Round 1
Reviewer 1 Report
population target is not clear in the abstract and in the discussion. I felt that this paper is better to separate into two papers. I felt that the information was abundant and complex, and that the goals were unclear and written in many of the previous papers. No new information has been added to the science
Author Response
Reviewer 1
Comment #1
“population target is not clear in the abstract and in the discussion”.
Answer
The authors really appreciate the reviewer’s comment. A new a paragraph was added at the abstract and at the end of the introduction highlightening the population target.
Revised Text:
Abstract:
This analytical review studies the nutrition situation of the EMR during the past 20 years by di-viding the region into four groups based on their income level - the low-income group (Afghani-stan, Somalia, Sudan, Syria and Yemen), the lower-middle-income group (Djibouti, Egypt, Iran, Morocco, Pakistan, Palestine, and Tunisia), the upper-middle-income group (Iraq, Jordan, Leba-non, and Libya) and the high-income group (Bahrain, Kuwait, Oman, Qatar, Saudi Arabia, and United Arab Emirates) - and by comparing and describing the estimates of the most important nutrition indicators, including stunting, wasting, overweight, obesity, anaemia, and early initia-tion and exclusive breastfeeding.
Introduction:
Therefore, this analytical review was conducted to clarify the nutrition status among children under five, children and adolescences from 5-19 years, adults and women at reproductive age in the different EMR income groups according to the latest available estimates. This is to investigate the correlations of these nutrition indicators with the income level, as well as to discuss the potential causes of the nutrition problems and describe the current challenges and progress regarding nutrition policies and programmes that could influence the life and well-being of EMR population.
Comment #2
I felt that this paper is better to separate into two papers.
Answer
The authors agree with the reviewer’s recommendation; the following sections have been removed “3.7. Micronutrients, 3.7.1. Vitamin A and 3.7.2. Iodine” as well as all the related parts in the discussion, also the following sections have been summarized “4.3.2. Persistent malnutrition exacerbated by local and global conflicts and crises and 4.3.3. The impact of COVID-19 pandemic and Ukraine’s war on food security and nutrition”
Comment #3
“I felt that the information was abundant and complex and that the goals were unclear and written in many of the previous papers. No new information has been added to the science”
Answer
The authors sincerely appreciate the reviewer’s comment regarding that the information is abundant and somewhat complex but the goal was to collect a comprehensive nutrition profile for the different EMR income groups. The data included in this paper are New data that have been calculated for the first time, after grouping the countries of EMR into 4 groups based on their income level and producing new estimates for each of these 4 groups by calculating weighted averages for each group (based on the country estimates and populations), and these estimates have been analyzed and described. Also, the potential reasons for the results have been discussed thoroughly. Moreover, the authors have consolidated information regarding the availability of nutrition policies and programmes in the countries of the region, which have not been done elsewhere. All of this information is important for the member states, nutrition specialists, policymakers to give them a comprehensive review of the current status and progress of nutrition indicators and programs in the region.
Reviewer 2 Report
Thank you for this very complex article, which will be an important source of information for many scientist working in public health nutrition.
However, due to its high complexity, the article needs to facilitate the access to its content for the interested reader, so I think you need to add:
- an enumeration of countries included. Yes, they are presented further in the article, in the classification of the WB, but please do it from the start. I never ever had imagined that Afganistan or Pakistan are EMR countries...
- provide a graphical or tabular presentation of the elements followed in the article (malnutrition, iodine, breastfeeding, etc). They are in the body of the article, but they are really drown in the sea of words. As a matter of fact, many elements presented would benefit of some graphical representations, some kind of graphical conclusions. Indeed, the article is already very long (from my opinion, there are too many concepts analyzed in one article), at least try to make it easy to understand.
Would you consider to split the article in 2?! As it is, it resembles more to a WHO report and it is rather difficult to read.
Author Response
Reviewer 2
Comment #1
“The article needs to facilitate the access to its content for the interested reader, so I think you need to add:
- an enumeration of countries included. Yes, they are presented further in the article, in the classification of the WB, but please do it from the start. I never ever had imagined that Afganistan or Pakistan are EMR countries.
Answer
Authors greatly appreciates the reviewer’s comment, the enumeration was added in the abstract and it is included in the methodology section and a map (Fig 1) was added to the introduction.
Revised text
Abstract:
This analytical review studies the nutrition situation of the EMR during the past 20 years by dividing the region into four groups based on their income level - the low-income group (Afghani-stan, Somalia, Sudan, Syria and Yemen), the lower-middle-income group (Djibouti, Egypt, Iran, Morocco, Pakistan, Palestine, and Tunisia), the upper-middle-income group (Iraq, Jordan, Leba-non, and Libya) and the high-income group (Bahrain, Kuwait, Oman, Qatar, Saudi Arabia, and United Arab Emirates) - and by comparing and describing the estimates of the most important nutrition indicators, including stunting, wasting, overweight, obesity, anaemia, and early initiation and exclusive breastfeeding.
Comment #2
“provide a graphical or tabular presentation of the elements followed in the article (malnutrition, iodine, breastfeeding, etc). They are in the body of the article, but they are really drown in the sea of words. As a matter of fact, many elements presented would benefit of some graphical representations, some kind of graphical conclusions.”
Answer
Following the reviewer’s recommendation, a graphical presentation (Fig 2) of the items analyzed in the article has been performed and inserted in the methodology section.
Comment #3
“Indeed, the article is already very long (from my opinion, there are too many concepts analyzed in one article), at least try to make it easy to understand. Would you consider to split the article in 2?! As it is, it resembles more to a WHO report and it is rather difficult to read.”
Answer
The authors agree with the reviewer’s recommendation; the following sections have been removed “3.7. Micronutrients, 3.7.1. Vitamin A and 3.7.2. Iodine” as well as all the related parts in the discussion, also the following sections have been summarized “4.3.2. Persistent malnutrition exacerbated by local and global conflicts and crises and 4.3.3. The impact of COVID-19 pandemic and Ukraine’s war on food security and nutrition”
Round 2
Reviewer 2 Report
None